

# UCN, the ultracold neutron source - neutrons for particle physics

**Bernhard Lauss⋆ and Bertrand Blau**

Paul Scherrer Institut, CH-5232 Villigen PSI, Switzerland

⋆ bernhard.lauss@psi.ch

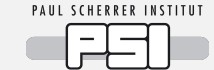

## Abstract

**Ultracold neutrons provide a unique tool for the study of neutron properties. An overview is given of the ultracold neutron (UCN) source at PSI, which produces the highest UCN intensities to fundamental physics experiments by exploiting the high intensity proton beam in combination with the high UCN yield in solid deuterium at a temperature of 5 K. We briefly list important fundamental physics results based on measurements with neutrons at PSI.**

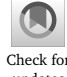

## 4.1 Introduction

Ultracold neutrons (UCNs) are at the lowest end of the neutron energy spectrum, with kinetic energies below about 300 neV, corresponding to velocities below 8 m/s, and to temperatures below 4 mK. Hence they are called "ultracold". This energy is the same as the neutron optical potential of certain materials. Thus material bottles can be used to store UCNs. This energy also corresponds to the potential difference of a neutron raised by 3 meters in the Earth's gravitational field, and also to the potential difference of a 5 Tesla magnetic field gradient acting on the neutron magnetic moment. Thus, UCNs can be relatively easily confined and manipulated. Therefore, they are a unique tool to study the properties of the neutron itself. The highest UCN intensities are needed to reach the highest sensitivity range in fundamental physics experiments; the most prominent such experiment is the search for a permanent electric dipole moment of the neutron (nEDM) [1, 2].

The idea to build an intense UCN source at PSI was formulated in the late 1990's. The UCN project was initiated and realized under the leadership of Manfred Daum. The technical design presented in 2000 [3, 4] was based on earlier studies in Russia [5–7] and a successful operation of a solid-deuterium based UCN source at the Los Alamos National Laboratory [8]. The main scientific goal was to push the sensitivity of the nEDM search to a new level. Several pioneering experiments by the PSI UCN group determining e.g. UCN production in solid deuterium [9,10] and UCN loss cross-sections [11, 12], paved the way for the final design. The UCN source was then installed as the second spallation neutron source at the PSI high intensity proton facility

(HIPA). After a short test beam period at the end of 2010, the UCN source started regular operation in 2011 [13–15] providing UCNs to experiments at three beam ports.

## 4.2 UCN Source Setup

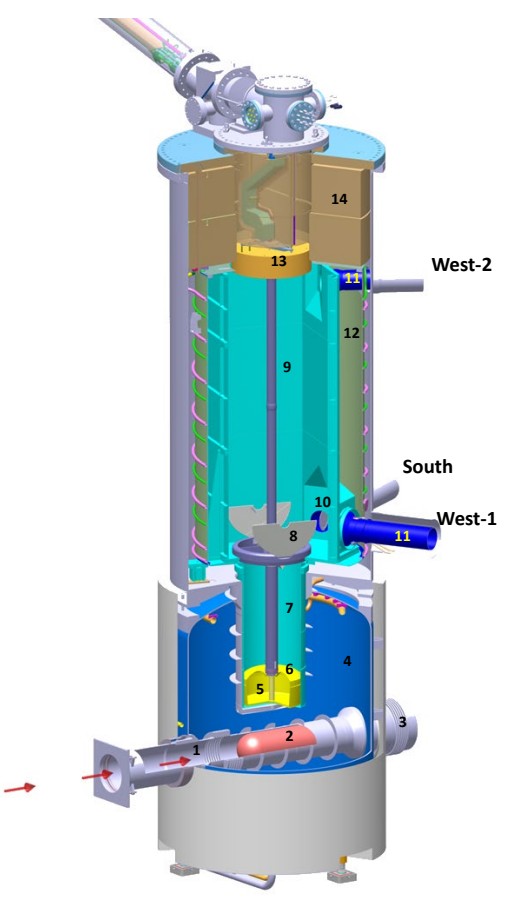

Figure 4.1: CAD image of the UCN tank with indicated parts relevant to UCN production and transport. 1 - proton beam tube, 2 - lead spallation target, 3 - target shielding, 4 - heavy water moderator tank, 5 - $D_2$ moderator vessel, 6 - lid, 7 - vertical guide, 8 - flapper valve, 9 - storage vessel, 10 - UCN guide shutter, 11 - UCN guide section, 12 - thermal shield, 13 - cryo-pump, at 5 K, 14 - iron shielding.

The PSI UCN source operates in the following way: The 590 MeV, 2.4 mA proton beam is deflected by a kicker magnet [16] for up to 8 s onto the lead spallation target (label 2) in Figure 4.1) [17]. In a spallation reaction between a lead nucleus and a 590 MeV proton, an average of 8 free neutrons is produced [18]. The neutrons are thermalized in the surrounding heavy water (label 4). The central moderator vessel (label 5) contains solid deuterium (sD$_2$) at a temperature of 5 K, which serves as both a cold moderator and as the UCN production medium. The cryogenics system needed for the manipulation, cooling and freezing of the deuterium [19] is shown in Figure 4.2. UCNs exit the moderator vessel through a thin aluminum lid (label 6 of Figure 4.1) into a vertical guide where the energy boost from the sD$_2$ surface [20] is lost by gravity. The flapper valve (label 8) of the 1.6 m$^3$ large storage vessel is closed at the end of the proton pulse. UCNs trapped in the storage vessel are delivered via about 8 m long neutron guides (label 11) to three beam ports, named West-1, South and West-2, with the latter extracting UCN from the top of the storage vessel. Great attention was spent

on quality checks of all elements, and extensive tests were performed before installation, e.g. the cryo-performance of several parts, most importantly the flapper valves (label 8) and UCN guide shutters (label 11). The UCN transport performance of all UCN guides [21] was confirmed prior to their installation. The overall neutron optics performance was later analyzed and understood in terms of a detailed simulation of the entire UCN source [22].

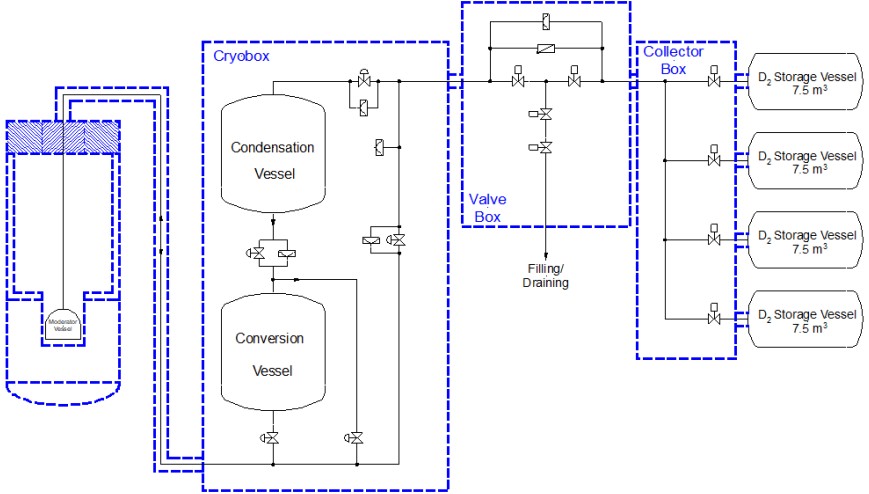

Figure 4.2: Schematic view of the subsystems needed for the preparation of the solid ortho-deuterium (see text).

The core of the UCN source is the solid deuterium moderator, which also serves as a UCN converter at a temperature of 5 K. The 30 liters of solid $D_2$ require very careful preparation in order to achieve optimal UCN output. A schematic view of the involved subsystem is shown in Figure 4.2. Preparation starts from the 30 $m^3$ ultra clean and isotopically pure $D_2$ gas, stored in large tanks at ambient temperature, which is slowly transferred by freezing into the 40 liter copper-made 'condensation' vessel. The $D_2$ is then slowly liquefied and transferred by gravity into the 'conversion' vessel at about 20 K where an ortho-D2 concentration of about 97% is achieved within 24 h by means of a spin-flip process on Oxisorb, a chromium-oxide-based catalyzer material. Raman spectroscopy is used to check the ortho-$D_2$ concentration [23, 24], which rises up to above 99% during longer operation periods. Once the required ortho concentration is reached in the conversion vessel, the liquid $D_2$ is transferred by gravity through a 10 m-long cold transfer line into the moderator vessel. Here it is slowly solidified over several days to achieve a good ice quality and, consequently, a high UCN output. The moderator vessel, shown in Figure 4.3a, is entirely made from AlMg3 with special coolant channels for the supercritical He cooling fluid at 4.7 K. These channels enter in the center of the vessel and direct the He stream to the outside wall, up and back in 8 separated sections, as schematically depicted in Fig. 4.3b.

The delivered UCN intensity reflects the quality of the achieved solid deuterium, likely a mosaic crystal with many defects and cracks, as was shown in the pioneering UCN experiments [9–12]. Slow freezing is crucial in the preparation process of the source. Figure 4.4 shows the typical UCN intensity behavior (green line) during such a slow freezing process. The vapor pressure (blue line) which is a direct measure of the $D_2$ (surface) temperature decreases from above 400 mbar (liquid $D_2$) to the triple point at about 171 mbar, where the liquid $D_2$ solidifies. After solidification the vapor pressure rapidly decreases below $10^{-2}$ mbar. The UCN output shows the opposite behavior. UCN loss processes dominate at higher temperatures, especially in the liquid $D_2$ and the high-density vapor located above the $D_2$. Once 5 K are

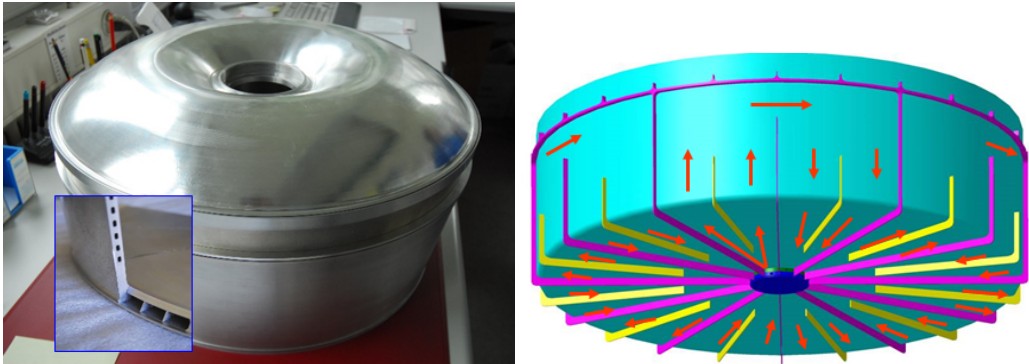

Figure 4.3: a) View of the moderator vessel with a cut insert view from a test vessel wall. b) Schematic view of the He coolant flow in the vessel demonstrating the sectional cooling.

reached, thermal losses are minimized and the UCN output is at its maximum.

## 4.3 UCN Source Performance

An important performance parameter is the number of UCNs delivered at a beam port in a given time interval as this determines the number of UCNs available in an experiment. The typical time structure of UCNs for a proton beam pulse is shown in Figure 4.5. The flaps of the storage vessel open before the proton beam hits the spallation target and their closing time is optimized with respect to the end of the proton pulse to provide the maximum number of UCNs to the experiments. The measured exponential decay of the UCN count rate at the West-1 beam port, Figure 4.5a), has a time constant of about 30 s, reflecting the emptying time of the central storage vessel through the West-1 guide into the UCN detector. The UCN rate at the South beam port behaves identically. If all shutters to the UCN guides remain closed on the storage vessel, the storage time constant for UCNs trapped inside the vessel is about 90 s. At the end of the filling/extraction period, which is typically 300 s long, the flaps are re-opened to be ready for the next proton beam pulse.

Figure 4.5b shows the UCN rate observed at the West-2 beam port located 230 cm above the bottom of the storage vessel [22]. The faster exponential decay demonstrates that the UCNs with energies high enough to reach up to 230 cm, are quickly drained through that port. The total number of UCNs delivered at the West-1 or South beam port was has been up to 45 million at the best operating conditions. The total number of delivered UCNs depends on the status of the solid deuterium, and was increased over the years with improvements in the operating conditions.

Several studies to understand all aspects of the UCN source have been conducted since its inauguration. The proton beam current and position is constantly monitored online with beam monitors. Neutron production and thermalization were checked using neutron activation measurements on gold. The observed activation was well reproduced in detailed neutron transport simulations using MCNP [18]. Neutron moderation was studied using tritium production in the solid $D_2$ moderator [24]. The high ortho $D_2$ concentration and the high isotopic purity of 0.09% H atoms (bound in HD molecules) of the $D_2$ was confirmed [24].

UCN transport from production in the solid deuterium to a beam port has been carefully studied as is detailed in the thesis works [25, 26]. Many geometry details were put into a full simulation model and the simulation results then matched well with observations [22].

The measured integral UCN intensity per beam pulse also shows a time dependence on

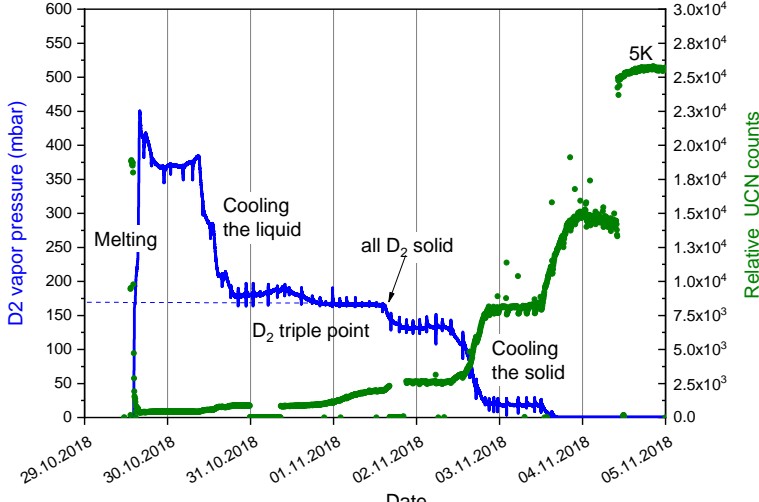

Figure 4.4: The observed behavior during the slow freezing of the deuterium. The vapor pressure of the $D_2$ (blue line) indicates the $D_2$ temperature. The $D_2$ was fully melted. When it reaches about 400 mbar vapor pressure, cooling starts and the $D_2$ slowly approaches the triple point at 171 mbar (horizontal dashed line). Here the $D_2$ solidifies. When the solid $D_2$ is further cooled down to 5 K the vapor pressure drops well below $10^{-2}$ mbar. The large increase in UCN output shown by the green bullets demonstrates the strong reduction in UCN losses within the $D_2$.

the scale of several hours to days, which considerably decreases the average UCN output. After several studies, a temperature-cycling procedure, called "conditioning", was developed that gets rid of the accumulated losses and regains maximum UCN intensity. This UCN count rate behavior is shown in Figure 4.6a, where the times when the conditioning procedure was applied are labeled by the vertical arrows. Figure 4.6b shows the measured deuterium vapor pressure in the moderator vessel during a 2-hour conditioning process. The rise in vapor pressure during a proton beam pulse, noted with the blue arrow, is minuscule. The rise during temperature cycling is up to about 50 Pa, depending on the total operation time since the previous conditioning. This is far below the triple-point pressure of 171 mbar and is due to sublimation, movement and resublimation of surface molecules during conditioning. Interesting enough, full rate recovery occurs.

One of the key characteristics of a UCN source is the UCN density that can be achieved in a given storage vessel. A stainless steel 'standard UCN storage vessel' with a volume of 20 liters [27] was built. This bottle was used to characterize the height-dependent UCN density at the West-1 beam port [26]. The UCN density peaks around 50 cm above the beam port as shown in Figure 4.7. This standard bottle was then used to characterize UCN densities of other sources in a comparable way [26, 28, 29]. As a result it has been shown that the PSI source provides world-leading performance to UCN storage experiments.

The PSI UCN source has been operating since 2011 on a regular schedule, mainly providing UCNs to the nEDM experiment. The yearly operation can be characterized by the integral of the proton beam current onto the UCN spallation target and the number of proton beam pulses, shown in Figure 4.8. The peak in 2016 was driven by the main data taking period of the nEDM experiment. The lower numbers in the subsequent years are due to longer periods of solid deuterium studies for UCN source improvements, which needed longer times with fewer

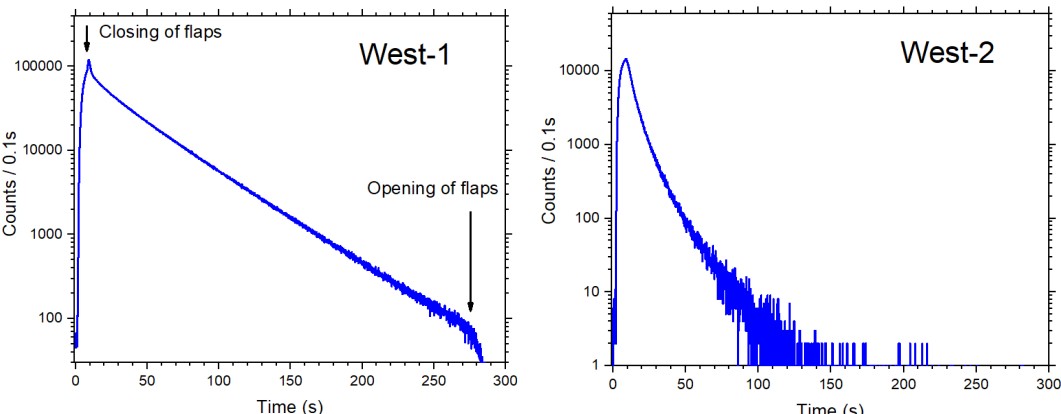

Figure 4.5: a) UCN counts after one proton beam pulse at the West-1 beam port. Closing and opening of the flaps refers to the central flapper valves. b) Same as a) but at the West-2 beam port.

proton beam pulses for performance checks.

## 4.4 Physics results at the UCN source

The construction of the UCN source at PSI was driven by the experiment to search for a neutron electric dipole moment. The resulting new nEDM limit was published in 2020 [2]. Significant physics results were also obtained on neutron properties and effects:

- a precision measurement of the mercury-to-neutron magnetic moment ratio [30];

- spin-echo spectroscopy with ultracold neutrons [31]

- measurement of gravitational depolarization of ultracold neutrons [32]

and on physics beyond the Standard Model:

- a limit for spin-dependent forces mediated by axion-like particles [33];

- the first laboratory limit for oscillating electric dipole moments [34];

- new limits for mirror-neutron oscillations in mirror magnetic fields [35].

Some of these results are treated in Section 18 [36] and Section 19 [37] of this volume.

## 4.5 Particle physics at the SINQ

The UCN source was conceived and built for research in fundamental neutron physics. However, the first spallation neutron source built at PSI was the SINQ facility [38]. While mainly dedicated to neutron scattering instruments, it has also been used as a polarized cold-neutron beam line for fundamental neutron physics. The 'FUNSPIN' beam line [39] (now called 'BOA') provided $6 \times 10^8$ neutrons $cm^{-2}s^{-1}mA^{-1}$ with 95% polarization [40].

The main physics results came from a series of measurements by the nTRV collaboration of neutron decay parameters. A precise determination of electron-neutron correlation coefficients $R$ and $N$ provided a precise test of the Standard Model and a search for exotic scalar and tensor interactions in neutron decay [41–44].

Another experiment produced a new measurement of the spin-dependent doublet neutron-deuteron scattering length [45, 46]. A Ramsey-type experiment resulted in an upper limit

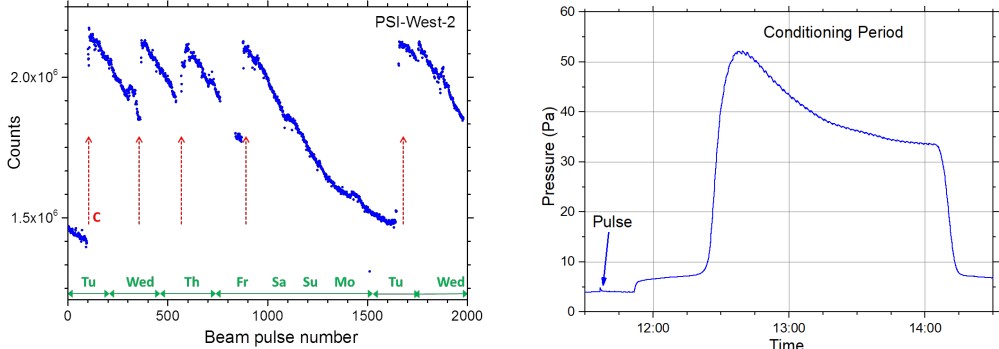

Figure 4.6: a) The UCN count rate behavior as observed over a 9-day operating period. The drop is interpreted as a frost effect. The count rate increases to the original level when the conditioning procedure is applied at the times depicted by the dashed arrows. Figure from [22] with kind permission of The European Physical Journal (EPJ). b) Vapor pressure of the solid deuterium surface in the moderator vessel during a full conditioning cycle.

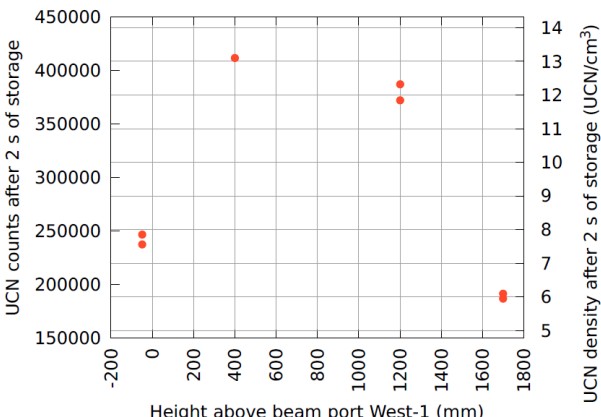

Figure 4.7: The UCN density measured at different heights with respect to the West-1 beam port. Picture from [26].

on the strength of an axial coupling constant for a new light spin 1 boson in the millimeter range [47].

Finally, we note the importance of the FUNSPIN beamline for many measurements conducted in preparation of the UCN source where many parameters of UCN production and loss were determined [9–12, 48–50].

## 4.6 Summary

A high-intensity source for ultracold neutrons, designed and built at PSI, has been operating since 2011. The layout, operation and performance are described. Some observations on the solid deuterium converter and its surface conditions are presented. Finally, a list of physics results in fundamental neutron physics results achieved with the UCN source and SINQ is given.

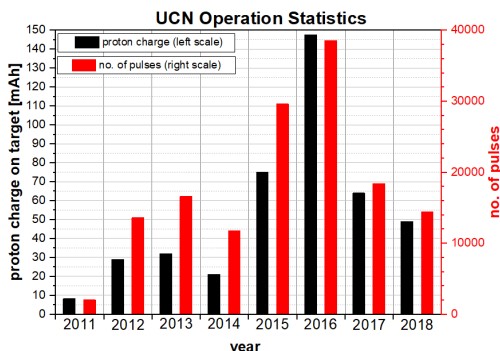

Figure 4.8: Annual statistics of the first operating years of the UCN source showing total accumulated beam current on target (black bars) and number of beam pulses (red bars) on the UCN spallation target.

**Acknowledgments**

Building the UCN source at PSI required the dedicated long-term support by many individuals and support groups at PSI that worked for several years together within the UCN source team. We especially acknowledge the invaluable contributions of all former and present members of the BSQ group and the UCN physics group.

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
