# Peer review of "UCN, the ultracold neutron source -- neutrons for particle physics"

_SciPost Physics Proceedings, doi:SciPost Phys. Proc. 5, 004 (2021)_

## Round 1 · Referee Report · Adrian Signer (Referee 1) · 2021-4-22

Report
We (the editors Cy Hoffman, Klaus Kirch, Adrian Signer) had the
opportunity to review an earlier draft of the article and were in
communication with the authors before the submission. All our comments
and suggestions have been taken into account. Hence, we think the
paper can now be published in the current form.
opportunity to review an earlier draft of the article and were in
communication with the authors before the submission. All our comments
and suggestions have been taken into account. Hence, we think the
paper can now be published in the current form.
Author: Bernhard Lauss on 2021-06-06 [id 1491]
(in reply to Report 2 on 2021-05-16)Thank you for the comments. The value of the isotopic purity (0.09%)was added. Section 4.5 is kept. This was part of teh original request to show that neutron particle physics at PSI is not limited to UCN.

---

## Round 1 · Referee Report · Anonymous (Referee 2) · 2021-5-16

Report
The manuscript by B. Blau, "UCN, the ultracold neutron source - neutrons for particle physics," provides an excellent description and overview of the PSI UCN source. As this source is one of the most intense and productive UCN sources in the world the paper will serve as valuable resource for those in the field of neutron research. Particularly noteworthy the very complete set of references describing the extensive literature that emerged during the development of the PSI source.
I have only two minor questions, neither of which raises to the level of a significant concern. In the otherwise complete description of the properties of the target, the isotopic purity of the the D_2 was described only as being “isotopically pure.” It might be helpful to include the actual purity as that is a quantity of considerable interest similar to the ortho-para fraction.
The second comment concerns the inclusion of section 4.5 which appears to be almost an afterthought. While its inclusion does not diminish the overall value of the article, I believe the paper would be improved by either dropping this section, and limiting the discussion to UCN or expanding it to the point where it include technical details comparable to those in the discussion of the UCN source.
I include these comments only as suggestions. Whether or not they are accepted does not affect my recommendation that this this article be published.
I have only two minor questions, neither of which raises to the level of a significant concern. In the otherwise complete description of the properties of the target, the isotopic purity of the the D_2 was described only as being “isotopically pure.” It might be helpful to include the actual purity as that is a quantity of considerable interest similar to the ortho-para fraction.
The second comment concerns the inclusion of section 4.5 which appears to be almost an afterthought. While its inclusion does not diminish the overall value of the article, I believe the paper would be improved by either dropping this section, and limiting the discussion to UCN or expanding it to the point where it include technical details comparable to those in the discussion of the UCN source.
I include these comments only as suggestions. Whether or not they are accepted does not affect my recommendation that this this article be published.

---

## Round 2 · List of Changes

The value of the iotopic purity of D2 was added.

---

## Editorial Decision

published